

# Impact of magnetic storms on the global TEC distribution

Donat V. Blagoveshchensky[1], Olga A. Maltseva[2], Maria A. Sergeeva[3,4]

[1]Saint-Petersburg State University of Aerospace Instrumentation, 67, Bolshaya Morskaya, Saint-Petersburg, 190000, Russia
[2]Institute for Physics, Southern Federal University, Stachki, 194, Rostov-on-Don, 344090, Russia
[3]SCiESMEX, LANCE, Instituto de Geofisica, Unidad Michoacan, Universidad Nacional Autonoma de Mexico, Morelia, 58089, Mexico
[4]CONACYT, Instituto de Geofisica, Unidad Michoacan, Universidad Nacional Autonoma de Mexico, Morelia, 58089, Mexico

*Correspondence to*: Maria A. Sergeeva (maria.a.sergeeva@gmail.com)

**Abstract.** The study is focused on the analysis of Total Electron Content (TEC) variations during six geomagnetic storms of different intensity: from Dstmin = – 46 nT to Dstmin = -223 nT. The values of TEC deviations from its 27-day median value (δTEC) were calculated during the periods of the storms along three meridians: American, Euro-African and Asian-Australian. The following results were obtained. For the majority of the storms almost simultaneous occurrence of δTEC maximums was observed along the Asian-Australian and Euro-African meridians at the beginning of the storm. The
transition from weak storm to superstorm (the increase of magnetic activity) almost does not affect the intensity of δTEC maximum. The effect revealed for the American sector during two storms was the movement of the disturbance front from Northern and Southern high latitudes towards the equator with the average velocity of ~ 400 m/s. The seasonal effect was most pronounced at Asian-Australian meridian, less often at Euro-African meridian and was not revealed at American meridian. Sometimes the seasonal effect can penetrate to the opposite hemisphere. The character of averaged δTEC
variations for the intense storms was confirmed by GOES satellite data. The behaviour of correlation coefficient (R) between δTEC at three meridians was analyzed for each storm. In general, R>0.5 between δTEC averaged along each meridian. This result is new. The possible reasons for the exceptions (when R < 0.5) were provided: time-shift of δTEC maximum at different latitudes along the American meridian, the complexity of phenomena during the intense storms and discordance in local time of geomagnetic storm beginning at different meridians. Notwithstanding the complex dependence of R on the
intensity of magnetic disturbance, in general R decreased with the growth of storm intensity.

**Keywords:** ionospheric disturbances, magnetosphere-ionosphere interactions.



# 1 Introduction

The changes in the Earth's geomagnetic field provoked by Space Weather events can cause ionospheric disturbances. The last are very complex phenomena. One of the parameters that help to estimate the ionosphere state change is the vertical Total Electron Content (TEC) that is the quantity of electrons in a column of unit cross section (Davies and Hartmann, 1997; Afraimovich and Perevalova, 2006). Usually, TEC is calculated using phase and code delays of GNSS satellites signals received by dual frequency ground-receivers. The ionosphere is represented by a thin shell of zero thickness at the altitudes of the ionospheric F-region when calculating TEC (Shaer et al., 1995; Komjathy, 1997). Though TEC is an integral characteristics (Electron content from the satellite to the ground), it is assumed that it characterizes the state of F-region of the ionosphere. This is due to the fact that the main contribution to electron content is provided by the ionospheric F-region. In recent years, TEC has been widely used for ionosphere diagnostics for local regions and on a global scale due to availability of signals in all-time, all-weather conditions around the globe (Panda et al., 2014) and the large coverage of GNSS receivers worldwide in comparison to other ground-based instruments such as ionosonde networks, radars, etc. Despite a large number of publications dedicated to the disturbed ionospheric state, new data are still interesting to analyze. In the majority of works data of vertical ionospheric sounding and TEC are used together. However, at present, TEC acts as an independent parameter, in particular to estimate disturbances as, for example, in works (Jakowski et al., 2006; Gulyaeva and Stanislawska, 2008).

The choice of events for the analysis usually varies from several storms, for instance 15 cases during 2006-2007 (Cander and Ciraolo, 2010) or 217 events between 2001 and 2015 (Liu et al., 2017), to the detailed studies of a particular event, as in (Astafyeva et al., 2015). In the present work we study the global ionospheric responses to six geomagnetic storms using TEC data. The storms of different intensity (from weak to severe) were chosen within a short time interval (one-year period). The effects of the storms of different intensity on ionosphere were compared.

A number of works addressed global ionosphere variations during disturbances. One of the possible approaches is to study the behaviour of parameters along different meridians (Mansilla, 2011; Astafyeva et al., 2015). The majority of studies of latitudinal or longitudinal dependences of ionospheric responses are limited to some latitude-longitude region, although there are studies of global density distributions. For example, Zhao et al. (2007) suggested the presence of a longitudinal effect of the ionospheric storm caused by geomagnetic disturbance. Rajesh et al. (2016) showed using GIM that mid-latitude electron density enhancements exhibit significant longitudinal dependence. Longitudinal varieties of the ion total density in the equatorial and mid-low latitudinal topside ionosphere at four local times were studied by (Chen et al., 2015). Latitudinal variations between longitudes 40ºE and 100ºE in the Indian zone were addressed by Bhuyan et al. (2002). Nogueira et al. (2013) examined the four-peaked structure in the observed topside ion density and its manifestation as longitudinal structures in TEC over South America. Dmitriev et al. (2013) performed the longitudinal analysis of the day-side ionospheric storms within the region of equatorial ionization anomaly during recurrent geomagnetic storms. Longitudinal features of electron





density distributions were studied in (Klimenko et al., 2015; Klimenko et al., 2016) for minimum solar activity using modeling, GPS and satellite observations.

The present study addresses the global longitudinal TEC features not limited by one particular latitude-longitude zone. Three longitude sectors being rather far from each other were chosen for the analysis: along the American meridian (100ºW), along the Euro-African meridian (15ºE) and along the Asian-Australian meridian (115ºE). The effects were studied along these three longitudes within the latitude interval between 60ºN and 60ºS.

The storms considered in the present study were also the object of several case studies mostly for some particular region. For example, Polekh et al. (2016) addressed the event of March 17, 2015; Astafyeva et al. (2016) studied ionosphere during June 22, 2015; Chashei et al. (2016) considered ionospheric effects during the storm on December 20, 2015, etc. In our case the focus is on global effects.

The aim of this work was to reveal the features of TEC variations during the particular geomagnetic storms along three meridians: American, Euro-African and Asian-Australian. The tasks were to: (1) obtain TEC variations along each meridian, (2) find if there is any correlation between these variations, (3) reveal if there is a peculiar character of TEC behaviour during the considered storms if compare to the quiet conditions and how this character depends on the intensity of disturbance and on the meridian itself.

## 2 Data used for the analysis

### 2.1 Parameters of magnetic storms

Six geomagnetic storms within one-year interval between March 2015 and March 2016 were chosen for the analysis. This period lays on the descending phase of solar activity cycle, not far from its maximum occurred in 2014. The majority of the storms occurred during the winter time in Northern Hemisphere (if categorize March as a winter month) and summer time in Southern Hemisphere. We have chosen the storms of different intensity. Figure 1 illustrates Dst-index variations characterizing the disturbances.

Table 1 provides the information about each event under analysis. The number assigned to each storm is given in the first column. The same numbers are used to label the panels of Figure 1. The dates of disturbances are given in the second column. The time moments of the beginning of the main phase of the storm (To) are given in the third column. Here, "o" means onset. Minimal Dst-index values and its corresponding time are indicated in the fourth column. The last fifth column presents the time moments of the end of the main phase of the storm (Te). Here, "e" means end. To moment was defined as a drastic Dst-index decrease as a result of the main phase development. Te moment corresponded to the end of the main phase when Dst value was about $(-10 \div -15)$ nT. The geomagnetic storms are presented in Table 1 from the less intense (first line) to the most intense (sixth line) according to the Dst-index. Gonzalez et al. (1994) introduced storm classification: intense storms are characterized by $Dst \leq -100$ nT, moderate storms - by $-100$ nT $\leq Dst \leq -50$ nT, weak storms - by $-50$ nT $\leq Dst \leq -30$ nT. According to this classification, the storm #1 (14.12.2015) is weak, the storm #2 (06.03.2016) is moderate,



the storms #3, #4, #5 and #6 are intense. The last storm (17.03.2015) is called a superstorm in literature because it was the most intense storm of solar cycle 24. Thus, all six considered storms are of different intensities.

## 2.2 TEC data

TEC values were obtained from Global Ionospheric Maps (GIM) produced by International GNSS Service (IGS).
GIM TEC are independently computed by four Analysis Centers of the International GPS Service for Geodynamics (CODE, JPL, UPS, ESA) and then ranked and combined according to the corresponding weight by the International GNSS Service to produce the IGS global vertical TEC maps (Hernandez-Pajares et al., 2009). These final IGS maps were used for this study. TEC values were extracted from IONEX-files, freely available by following the link ftp://cddis.gsfc.nasa.gov/pub/gps/products/ionex. GIM provides the spatial resolution of 5º longitude and 2.5º latitude
worldwide, thus it is a useful tool for ionosphere diagnostics on a global scale.

For each observation point median TEC value was calculated on the basis of 27 previous days for every two hours of the day (UT). Thus, its own median value was obtained for each day every two hours. Furthermore, the deviation of TEC was calculated and plotted during each storm as well as six days before and six days after it following Eq. (1):

$$\delta\text{TEC} = \frac{(\text{TECobs} - \text{TECmed27})}{\text{TECmed27}} \times 100\% ,\qquad\qquad (1)$$

where TECobs is the observed absolute value, TECmed27 is a median value calculated for the 27 days prior to the day of observation.

## 2.3 Satellite and geomagnetic data

Data from GOES weather satellites that circle the Earth in a geosynchronous orbit was used in the analysis (https://satdat.ngdc.noaa.gov/sem/goes/). The altitude of their orbit is about 35800 km. GOES-13 is positioned at 75ºW
longitude and the equator monitoring North and South America and the Atlantic Ocean. GOES-15 is positioned at 135ºW longitude and the equator monitoring North America and the Pacific Ocean. The coverage by two satellites extends approximately from 20ºW longitude to 165ºE longitude. The instruments for near-Earth Space Weather monitoring are installed on board including magnetometer, X-ray sensor, high energy proton and alpha detector, and energetic particles sensor.
To estimate geomagnetic conditions, the Dst-index values were used. This index is an indicator of global Space Weather effects. Data is freely available by following the link http://wdc.kugi.kyoto-u.ac.jp/dstdir/index.html.





## 3 Discussion of results

### 3.1 Specific features of TEC variations during the considered storms

Variations of δTEC were the main source of information about the changes in the ionosphere. According to this data, the bursts of δTEC occurred at the beginning of magnetic disturbance, between the moments To and Te. The duration of these bursts varied within several hours. The behaviour of δTEC along American, Euro-African and Asian-Australian meridians was studied with 10º step in latitude from 60ºN to 60ºS.

### 3.1.1 Weak δTEC variations

Sometimes manifestations of disturbance in TEC during geomagnetic storms were weak or absent within the latitude range of ±20º near the equator. Figure 2 provides the example for the storm of December 31, 2015 at the Euro-African sector. Here, for the economy of space the plots are shown with the 20º latitude step along the longitude. Days in Universal Time (UT) were laid off along the X-axis; additionally markings were laid every 2 hours (UT).

### 3.1.2 Seasonal effect

The presence of seasonal effects in δTEC variations was revealed for the following cases.

(a) During the storm #2 (March 6th, 2016) the positive phase of disturbance was the dominant effect in δTEC variations during the night hours (UT) between March 6-7 along the Asian-Australian meridian from latitude 60ºN to latitude 0º. In contrast, at the same meridian from 10ºS to 60ºS the positive phase was followed by negative phase. In other words, during this storm the positive disturbance covered the latitudes of winter hemisphere, meanwhile summer hemisphere was characterised by positive disturbance followed by negative disturbance.

(b) Similar picture was observed along the same (Asian-Australian) meridian during the storm #4 (December 20th, 2015). However, though the general tendency of δTEC was similar along the whole meridian (increase of values followed by decrease), in terms of phases the positive phase followed by decrease of values prevailed in Northern (winter) hemisphere from latitude 60ºN to 30ºN (Fig. 3 panel a). Further, from 20ºN to 60ºS, the δTEC increase was less pronounced and was followed by the clear negative phase. Here, the "summer" effect penetrated into the "winter" hemisphere.

(c) During the same storm #4 along the Euro-African meridian from December 20th to December 22nd (0 UT) the disturbance showed the "positive-negative-positive" sequence of phases from 60ºN to 10ºN. Here, the second positive phase was much more intense and the whole disturbance within the interval 30ºN - 0º began earlier. The latitudes of Southern hemisphere 0º- 60ºS were covered by the negative phase during December 21st with preceding positive phase almost disappearing.

(d) During the storm #5 (June 23, 2015) along the Euro-African meridian the negative phase in the form of two bays was observed from 60ºN to 0º (Fig. 3 panel b). From 10ºS to 60ºS the disturbance had more complex character and included





two or more positive phases. At the same time along the Asian-Australian meridian the negative phase was observed between 60ºN and 20ºN (Fig. 3 panel c). Starting from 10ºN positive phase (sometimes various peaks) was followed by negative phase. At that, the positive phase was in the form of a very intense burst (+ 180% and more) at latitudes between 20ºS and 60ºS. In this case, the "winter" effect penetrated into Northern Hemisphere from South.

To sum up, according to our data (cases (a)–(d)), the seasonal effect consists in general dominance of negative phase (decrease of TEC) in summer and positive phase (increase of TEC) in winter. This conclusion is in accordance with the case study (Kil et al., 2003). In the present study the effect was observed mostly over the Asian-Australian sector and no seasonal effect was registered over the American sector. Kil et al. (2003) addressed the case of magnetic storm of July 20[th], 2000, using GIM and low-orbit satellite data. They revealed clear seasonal effects: a dominance of the negative ionospheric

storm in the summer (northern) hemisphere and the pronounced positive ionospheric storm in the winter (southern) hemisphere. Kil et al. (2003) also found that the Northern "summer" negative phase penetrated into the Southern hemisphere. Our results also prove the possibility of penetrating of the seasonal effect to the opposite hemisphere. However, in our case both examples (b) and (d) showed such penetration from Southern to Northern Hemisphere: summer effects and winter effects respectively. Thus, we may conclude that it does not depend on the season itself or on the hemisphere.

The storm analyzed by (Kil et al., 2003) was very intense (Dstmin = -300nT). Our examples prove that the seasonal effect can be observed during the magnetic disturbance of less intensity (but still intense): -98 nT (a), -155 nT (b and c), -204 nT (d).

Here, we briefly mention that Zhao et al. (2007) also showed with GIM TEC that during magnetic disturbances a negative phase occurred with higher probability in the summer hemisphere, while a positive phase - in the winter

hemisphere. According to these authors, negative phase was most prominent near geomagnetic poles and positive phase was far from polar regions. According to our data within the latitudes ±60°, the positive phase is very probable during the disturbances. At the same time it is not contradictory as each geomagnetic storm is a particular unique event.

To conclude, the seasonal effects had longitudinal dependence: observed mostly over the Asian-Australian sector, sometimes over Euro-African sector and no seasonal effect was registered over the American sector.

### 25  3.1.3 Features of δTEC variations along the American meridian

Figure 4 illustrates the example of maximal δTEC bursts along the American meridian during the same storm as in Fig. 2 (in the middle of each panel of the figure). Left panels display variations in the Northern Hemisphere, right panels – in the Southern Hemisphere. The latitude step of 20º is chosen for space saving. The effect of δTEC bursts was observed at all latitudes from 60ºN to 60ºS. The following feature was revealed (Fig.4): the gradual shift of the δTEC maximum occurrence

in time is seen from latitude 60ºN to latitude 0º and from 60ºS to 0º. This means that the disturbance front moved from northern and southern high latitudes towards the equator. It is possible to estimate the velocity of this disturbance. The distance along the Earth's surface between the latitudes 40ºN and 0º or between 40ºS and 0º is approximately 40*111 = 4440 km. The time-shift is about 3 hours along latitude. Consequently, the approximate average velocity of the disturbance front





movement was about 1480 km/h or 400m/s, which confirms the existing understanding of the issue (Danilov, 2013 and references therein).

In Southern hemisphere during the summer (when the storm occurred) the background (solar-induced) thermospheric circulation is directed towards the equator all the time, thus helping the disturbance to propagate. In Northern hemisphere the picture seems to be more complex. It was winter. During the day hours the background circulation was directed polewards preventing the disturbance from moving lower and during the night it was directed equatorwards. It is proved by our data. The δTEC peak occurred at latitudes $60^oN-50^oN$ around 07 LT in the morning and then it was observed only at 19 LT at latitudes $40^oN-20^oN$ with amplitude being less at $20^oN$. Furthermore, it occurred around 23 LT at latitude $10^oN$ and between 23LT and 03 LT near the equator.

The case of similar scenario was observed during the storm December $20^{th}$, 2015. δTEC peak shift was registered again along the American meridian from latitudes $60^o$ towards the equator during approximately 12 hours. The effect was observed in both hemispheres. The disturbance front was probably moving from high latitudes towards the equator as in the previous example. As this effect was observed only along the American meridian, it may be supposed that it is related to the more "southern" location of the magnetic pole than at European or Asian meridians. Similar assumption was made in (Blagoveshchensky et al., 2003).

It worth noting that the behaviour of maximum bursts described for storms 31.12.2015 and 20.12.2015 is not characteristic for other storms and may be called unusual. For other storms and meridians almost simultaneous occurrence of δTEC peaks was observed at high northern and southern latitudes and at the equator along the same meridian. The last statement is proved in the following subsection.

### 3.1.4 Global picture of δTEC variations at three meridians

Figure 5 shows the averaged δTEC behaviour. Each panel (a-f) represents the results for the particular storm: from the weakest (panel a) to the strongest (panel f). Storm dates are indicated below the panels. The time-interval on the X-axis is the interval between To and Te (individual for each storm), according to Table 1. Each panel consists of three plots: upper plot represents variations in the American sector, middle plot – in Euro-African and the lower plot – in Asian-Australian sector. The curve on each plot represents δTEC values averaged along one meridian over the latitudes $60^oN – 60^oS$ with $10^o$ step (δTECav). In other words, the final δTECav curve represents the average of 13 δTEC values from different latitudes. This averaging is possible because according to our data the tendency of increasing or decreasing of δTEC was the same at different latitudes along one meridian in most cases (without the regard to the phase). The specific cases are described above and also considered below.

First, it is seen that the maximal δTECav lays close to To. Physically, it is explained by the fact that usually the drastic increase of particle flows from magnetosphere into ionosphere occurs at the beginning of each storm that, in turn, results in TEC disturbance. It is known, that during the development of disturbance the critical frequencies of ionosphere



decrease lower than their initial quiet level (Blagoveshchensky, 2011). The same behaviour is observed in TEC: minimum of δTECav values is observed after the increase of δTECav, caused by the main phase of storm.

The main feature seen in the panels "a", "b", "e" is approximately the same time (UT) of δTECav maximum occurrence at all the latitudes along three meridians. In regard to panels "c" and "d", their results were discussed above. To add, the δTECav maximum took place at the same time at Asian-Australian and Euro-African meridians. For American meridian the peaks are shifted in time as it was mentioned before and the peaks themselves are more diffused if compare with Asia and Europe. Let us consider a more detailed picture of each panel of Fig. 5.

Panel (a) has the shortest interval (To-Te) in consequence of the weakness of geomagnetic storm on December 14$^{th}$, 2015. This weak intensity is the reason of the slow ionospheric response and the particle precipitation occur with a certain delay from To moment. At that, the moments of δTEC maximums coincide at three meridians.

In panel (b) δTECav maximums were well-pronounced and coincided in time at three meridians during the moderate storm on March 6$^{th}$, 2016.

Panel (c) illustrates the results for the storm on December 31$^{st}$, 2015 which specific details were discussed above. Time of δTECav maximums occurrence was the same only at Asian-Australian and Euro-African meridians.

Panel (d) illustrates the picture similar to panel "c", but for the storm on December 20$^{th}$, 2015.

Panel (e) shows the results for the intense storm of June 23$^{rd}$, 2015. It was the only storm among the six that occurred during the summer at Northern Hemisphere and during the winter in Southern Hemisphere. However, no specific details were revealed in comparison to other considered storms.

Panel (f) shows the results for superstorm of March 17$^{th}$, 2015. Though it is the most intense storm among the six, in general δTECav variations do not differ from the other storms: the increase of δTECav was followed by its decrease. However, the negative phase was more pronounced if compare with the positive phase.

To conclude, there is no dependence of δTECav maximums at three meridians on the intensity of magnetic activity. We recall that the intensity of storms grows from panel "a" to panel "f", but no increase in δTECav variations is detected.

### 3.2 Data of GOES-13 satellite

To compliment the analysis of Figure 5 and for better understanding of phenomena the results of measurements at GOES satellite were involved in this study. Its orbit in the near Earth space is at the altitude of 35800 km that is in the Earth's magnetosphere. Among the measurements performed at the satellite there were the intensity of X-rays, protons with energies from >1 to >100MeV, electrons with energies from >0.8 to >4 MeV.

GOES data was studied during the periods of all six geomagnetic storms (Fig.1). The particle flows of protons and electrons were registered for all considered storms. However, for storms #1 - #4 (Fig.1, Table 1) the intensity of these flows did not differed significantly from its undisturbed rate. Rather high levels of particle flows were observed only for storms #5 and #6. For Dst values of order of -150 nT (storm #4) the flows level was rather low and only for Dst being lower than -200 nT it was significant (intense storms #5 and #6 with Dst values being -204 nT and -223 nT respectively). Thus, it was



impractical to consider satellite data for the first four storms #1 - #4. Figure 6 shows the flows variations for storms #5 and #6. The moments To and Te are labeled with vertical lines for both storms. Figure 1 shows that the amplitudes and the shapes of Dst curves were close for both disturbances. It was of interest to compare the satellite measurements of high energy particles - protons and electrons. Protons variations (p) are plotted in the upper half of the plots of Fig. 6, electrons variations (e) – in the lower parts. To moment for two storms was approximately at the moment of maximal proton radiation and the beginning of minimal electron flows. Then, the decrease of proton flow occurred in the interval To-Te, but electron flows increased from its minimal to maximal values during the same time. In general terms, the proton and electron flows during magnetic storms are probably not directly connected with electron density in the ionosphere (Afraimovich and Perevalova, 2006). However, implicitly it is possible. The increase in δTECav values (Fig.5) at the beginning of the storm was probably related to the maximum of proton rates. The decrease in electron flux coincided with δTECav decrease. Further, the drastic growth of electron flux intensity took place which led to δTECav growth in Fig.5. In particular, for the storm #5 (June 23rd, 2015) Fig. 5 illustrates δTECav bursts before June 23rd, then the decrease to the minimum around June 24th and then again some increase between June 24th – 25th. Similar picture was observed during storm #6 (March 17th, 2015): the maximal intensity of the proton flux was accompanied with δTECav small increase (not significant in this case but existing) near To moment (Fig.5,f) and then the decrease of the flux took place. During March 17th-18th the electron flux minimum was observed and then its increase. Thus, the character of δTECav behaviour for two storms in some way is proved by satellite data of energetic protons and electrons.

**3.3 Similarities and differences of δTEC response at different meridians during the storms**

We estimated a degree of correlation between δTECav at different meridians for each storm within the interval To-Te. This interval was different for each storm. Thus, 16 δTECav values were found within To-Te during storm #1; 23 values – during storm #2; 25 – during storm # 3; 49 - during storm # 4; 33 – during storm #5 and 58 – during storm #6. The distances in degrees between the meridians are the following: American – Euro-African (Am-E) – 115º, Euro-African – Asian-Australian (E-A) - 100º, Asian-Australian – American (A-Am) - 145º. The shortest distance is between E-A meridians and the largest – between A-Am meridians. Table 2 shows values of correlation coefficient (R) that was calculated between δTEC values at different meridians: (1) averaged at along the whole meridian (bold type), (2) averaged along the meridian in Northern Hemisphere (normal type), (3) averaged along the meridian in Southern Hemisphere (italic type).

**3.3.1 δTEC averaged along the whole meridians**

Table 2 illustrates the following features for averaging along the whole meridian (bold type).

- Rather high degree of correlation (R>0.5) took place between the δTEC variations during storms #1-#5 for all meridians except two values R = 0.148 and R = 0.430 between Asian-Australian and American meridians. This is explained by the time shift of δTEC peak along the American meridian as shown in Fig.5 (panels c and d). We associate low



correlations during storm #6 with the complexness of local phenomena because of the high intensity of the storm (including no correlation in the case A-Am).

- The highest R values (if comparing three pairs of meridians) were found between European and Asian-Australian sectors in five cases of six.

- The highest R values between all three meridians (R>0.5) were during the weakest storm #1. This corresponds to the physics of phenomena. Perturbations and irregularities in the ionosphere are more pronounced during intense disturbances than during moderate or weak disturbances. During the weak storm the ionosphere structure is not significantly changed and its global stability is retained.

- The lowest R values (in bold) took place between Asian-Australian and American sectors if compare to other two pairs at least for five storms of six. It is probably explained by the fact that the distance between the American and Asian meridians is the largest ($145^{o}$). Another possible cause is that To were found in the contrary local time zones (day or night local hours) for these two meridians during all storms under analysis.

- The not evident, mixed dependence of R on the intensity of magnetic disturbance is common for all three meridians. For example, the comparison of R for storms #1 - #4 shows that R are decreasing from values R = 0.884 (Am-E), R = 0.815 (E-A), R = 0.744 (A-Am) to values R = 0.522 (Am-E), R = 0.615 (E-A), R = 0.430 (A-Am). This is in accordance with physics of phenomena. However, the transition from the storm #4 to the storm #6 shows inverse dependence: some growth of R instead of its decrease for storm #5. Nevertheless, in general, R behaviour in dependence to the intensity of magnetic disturbance (transition from storm #1 to storm #6) showed the decrease of R values, which is to be expected. The lowest R values were for the most intense storm.

**3.3.2 δTEC averaged along meridians in each hemisphere**

It is known that TEC behaviour has a seasonal dependence (Afraimovich and Perevalova, 2006). As the seasons are opposites in two hemispheres, the effects in North and South can be different. In general, it is revealed that the intense bursts of δTEC took place at subpolar latitudes of both hemispheres. To compare "northern" and "southern" data first the averaging of δTEC was performed along each meridian separately in each hemisphere: between the latitudes 60ºN-10ºN (northern) and then between the latitudes 10ºS – 60ºS (southern). Though the averaging along the meridian implies only qualitative, not quantitative estimate of deviations, it was of interest to analyze the effects separately. Table 2 presents the results of R calculations made separately for Northern (normal type) and Southern (italic type) hemispheres.

- For two storms #5 and #6 close by their intensities of disturbance, but different by the season of occurrence (summer/winter and winter/summer) the following is characteristic. R<0.5 in Northern hemisphere (summer) and R>0.5 in Southern hemisphere (winter) at all three meridians during the storm #5. For the storm #6 the opposite picture is seen. R<0.5 in both hemispheres and there was no correlation in cases Am-E and A-Am. But in cases of correlation existence, R was lower in Southern hemisphere (summer) than in Northern hemisphere (winter) when the correlation was detected (E-A). It may be related to the seasonal effect, but more statistics is needed to conclude.



- Comparison of R for Southern and Northern hemispheres shows rather high degree of correlation in both hemispheres simultaneously (R>0.5) only for the weak storm #1. For other storms  the number of cases when R<0.5 increases with the disturbance intensity: one case for the storm #2, two cases for the storm #3, three cases for storms #4 and #5,  five cases for storm #6. In other words, the difference in R values in Northern and Southern hemispheres grows with the increase of magnetic activity. It results that seasonal effect has impact here.

- The correlation coefficients, R, calculated along a whole meridian (bold values) are close to maximal R values either of Northern or Southern hemisphere.

### 3.3.3 δTEC at three meridians in each latitude sector (without averaging)

We briefly mention that R behaviour was also studied without averaging (at each latitude separately). The results confirmed the last conclusion of issue 3.3.1: the lower the intensity of magnetic storm, the more the number of moderate and strong correlations between δTEC at different latitudes (R>0.45). Mild and weak correlations prevailed with the growth of the intensity of storms. The number of negative correlation also increased with the storm intensity growth. For instance, 11 such cases of total 39 were found for the superstorm #6.

For storm # 5 (June 23, 2015) R behaviour was found to be similar for all three pairs of meridians: R was positive within the latitudes ±60º and ±10º (in both hemispheres) and R was rather low or negative within the interval from 10Nº to 10Sº. Consequently, the ionosphere processes in equatorial zone were due to different physical causes at three meridians.

### 3.4 Conclusions

The features of behaviour of Total Electron Content deviation from its 27-day median value were studied during six geomagnetic storms of different intensity along three meridians: American, Euro-African and Asian-Australian. The storms were chosen within a short period of time (one year). Though six storms is not a big statistics, some features of TEC variations during these particular events were obtained.

1) During the majority of considered storms at Asian-Australian and Euro-African meridians the maximum of δTEC bursts occurred almost simultaneously at high latitudes in North and South and at the equator provided that the consideration was along each meridian separately. The specific effect was revealed at the American meridian during the storms of December 31st, 2015 and December 20th, 2015: the gradual shift of δTEC burst maximum from latitudes 60ºN and 60ºS towards the latitude 0º. This proves that the front of disturbance moved from Northern and Southern high latitudes to the equator. The average velocity of the front movement was about 400m/s. This value is close to obtained in earlier works. As this effect was observed only along the American meridian, it probably can be related to the more southern location of the magnetic pole, than at other two meridians (Euro-African and Asian-Australian).

2) It was revealed that the beginning of TEC disturbance during the superstorm March 17, 2015, qualitatively did not differ from the beginning of other storms: increase of δTECav was followed by its decrease. The transition from weak storm to superstorm (the increase of magnetic activity) almost does not influence the intensity of δTECav maximum.




3) The seasonal effect (general dominance of negative/ positive phase in summer/winter) was observed mostly at Asian-Australian meridian. No seasonal effect was registered over American sector. Our results prove the possibility of the seasonal effect penetrating to the opposite hemisphere (in our case from the Southern to Northern Hemisphere). We did not found proof of dependence of such penetrations on the season itself or on the hemisphere.

4) The character of δTEC for most intense storms under analysis (June 23$^{rd}$, 2015 with Dstmin = -204 nT and March 17$^{th}$, 2015 with Dstmin - -223 nT) is rather similar despite of the opposite seasons of occurrence of storms and in some way is confirmed by GOES satellite data of energetic proton and electron fluxes.

5) The analysis of correlation coefficients between averaged δTEC variations at three meridians during each storm within the interval To-Te showed the following.

- The degree of correlation between averaged along a whole meridian δTEC values at three meridians was rather high (R>0.5). This result is new. There are five exceptions of 18 cases from Table 2: (a) R = 0,148 and R = 0.430, both found between Asian-Australian and American meridians, and (b) low R during the most intense storm #6. Issue (a) is related to the time-shift of δTEC maximum at different latitudes along the American meridian. The reason of the shift is provided. Issue (b) is associated with the complexity of phenomena during the most intense storm.

- The highest coefficients of correlation between averaged along a whole meridian δTEC (all three R>0.5) took place during the weakest storm. This is due to the fact that during the weak storm the ionosphere structure is not significantly changed and its global stability is retained. Comparison of R between δTEC averaged separately in Northern and Southern hemispheres also showed that high degree of correlation for both hemispheres R>0.5 took place only for the weak storm. The difference between hemispheres increased with the increase of magnetic activity, that probably again is explained by seasonal effect.

- The lowest coefficients of correlation (through all the storms in general) were found between Asian-Australian and American meridians. The reasons may include the largest distance between these meridians and discordance in local time of To occurrence.

- The not evident, mixed dependence of R on the intensity of magnetic disturbance is common for all three meridians. Nonetheless, the transition from weak to the most intense storm shows the decrease of correlation rates to the point of absence or even negative correlations. This result is new. It is confirmed by correlation coefficients between both averaged δTEC and δTEC at each latitude separately. In general, the more the intensity of magnetic disturbance, the lower the correlation rates between δTEC variations at three meridians.

- Calculation of R separately for two hemispheres allowed us to reveal that the most intense δTEC bursts took place at subpolar latitudes of both hemispheres. For two storms 23.06.2015 and 17.03.2015 close by the intensity but different by the season the following is revealed. For summer storm 23.06.2015 R values were less than 0.5 in Northern hemisphere and more than 0.5 – in Southern hemisphere between all three meridians. For storm 17.03.2015 R values were less than 0.5, but in general, the picture was vice versa: correlation coefficients were lower in Southern hemisphere and higher – in Northern (when correlation was detected). The seasonal effect probably plays a main role here.



- For the storm of June 23, 2015, R between δTEC at each latitude for all three pairs of meridians was positive within the latitudes ±60º and ±10º (in both hemispheres) and was rather low or negative within the interval 10Nº-10Sº. Consequently, the ionosphere processes in equatorial zone were the subject of different physical causes at three meridians.

**3.5 Acknowledgments**

5      The work of Blagoveshchensky D.V. was supported by grant № 18-05-00343 from Russian Foundation for Basic Research. The work of Maltseva O.A. was supported by grant under the state task N3.9696.2017/8.9 from Ministry of Education and Science of Russia. SCiESMEX is partially funded by CONACyT-AEM Grant 2014- 01-247722, CONACyT LN 269195, and DGAPA-PAPIIT Grant IN106916.

The authors express their gratitude to the services of IGS for the opportunity of using IONEX data via Internet.

25



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

25

30



**Table 1.** **Characteristics of the geomagnetic storms used in the analysis.**

| # | Date of storm beginning | To | Dst min;      hour;      date | Te |
|---|---|---|---|---|
| 1 | 14.12.15 | 16 UT,   14.12.15 | -46 nT;     20 UT;     14.12.15 | 22 UT,    15.12.15 |
| 2 | 06.03.16 | 16 UT,   06.03.16 | -98 nT;     22 UT;     06.03.16 | 12 UT,    08.03.16 |
| 3 | 31.12.15 | 12UT,    31.12.15 | -110 nT;   01UT;     01.01.16 | 12 UT,    02.01.16 |
| 4 | 20.12.15 | 00 UT,   20.12.15 | -155 nT;    23UT;     20.12.15 | 24 UT,    23.12.15 |
| 5 | 23.06.15 | 13UT,    22.06.15 | -204 nT;    05UT;     23.06.15 | 06 UT,    25.06.15 |
| 6 | 17.03.15 | 06UT,    17.03.15 | -223 nT;    23UT;     17.03.15 | 24 UT,    21.03.15 |

25

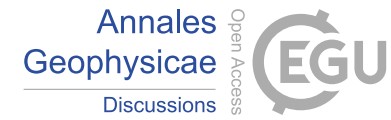

**Table 2. Correlation coefficients between δTEC at three meridians.**

| # | Date of storm | American - Euro-African (Am-E) | Euro-African - Asian-Australian (E-A) | Asian-Australian - American (A-Am) |
|---|---|---|---|---|
| 1 | 14.12.15 | **0.884** | **0.815** | **0.744** |
|   |          | 0.745 | 0.857 | 0.621 |
|   |          | *0.561* | *0.640* | *0.744* |
| 2 | 06. 03.16 | **0.737** | **0.689** | **0.791** |
|   |          | 0.746 | 0.298 | 0.577 |
|   |          | *0.635* | *0.673* | *0.758* |
| 3 | 31.12.15 | **0.644** | **0.791** | **0.148** |
|   |          | 0.685 | 0.738 | 0.574 |
|   |          | *0.394* | *0.808* | *0.012* |
| 4 | 20.12.15 | **0.522** | **0.615** | **0.430** |
|   |          | 0.556 | 0.499 | 0.729 |
|   |          | *0.239* | *0.508* | *0.128* |
| 5 | 23.06.15 | **0.672** | **0.832** | **0.724** |
|   |          | 0.449 | 0.158 | 0.467 |
|   |          | *0.717* | *0.854* | *0.716* |
| 6 | 17.03.15 | **0.362** | **0.463** | **0.004** |
|   |          | 0.279 | 0.172 | 0.332 |
|   |          | *0.071* | *0.509* | *-0.086* |




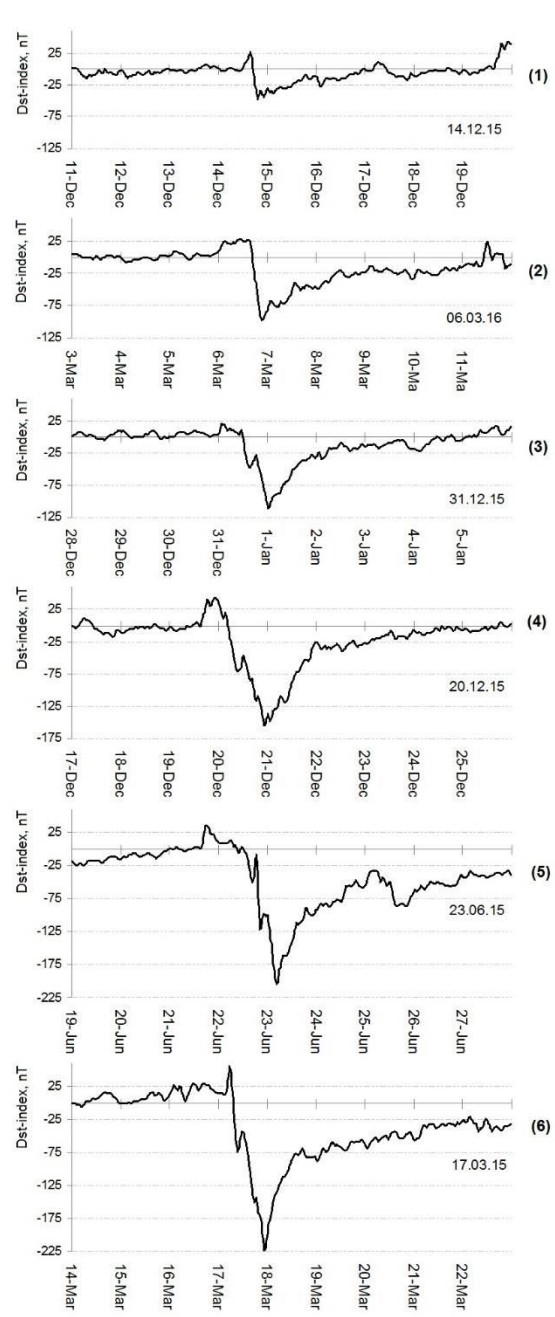

**Figure 1: Dst-index variations during the periods of six geomagnetic storms under analysis.**





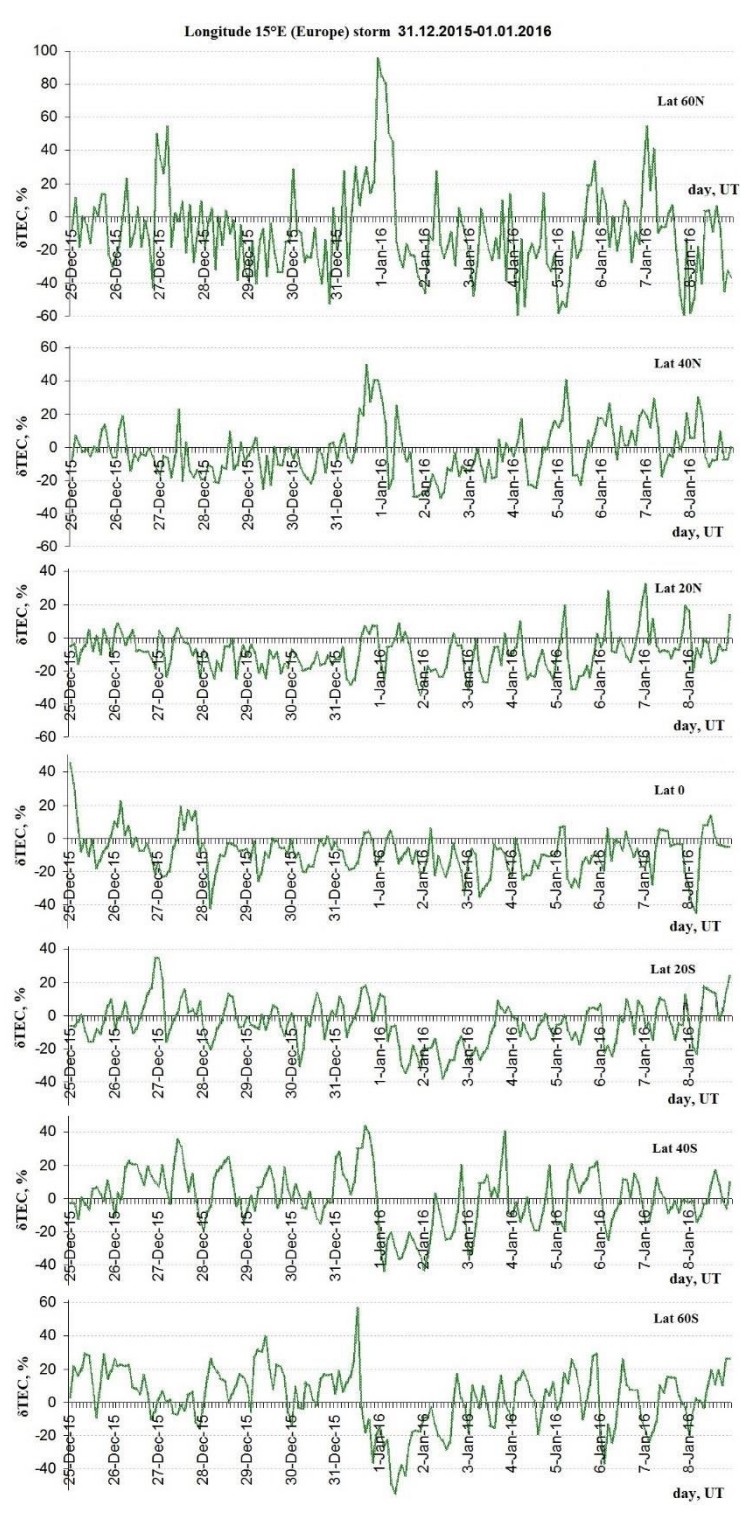

**Figure 2: Weak manifestation of TEC effects within the latitudes ±20° during the storm of December 31st, 2015.**





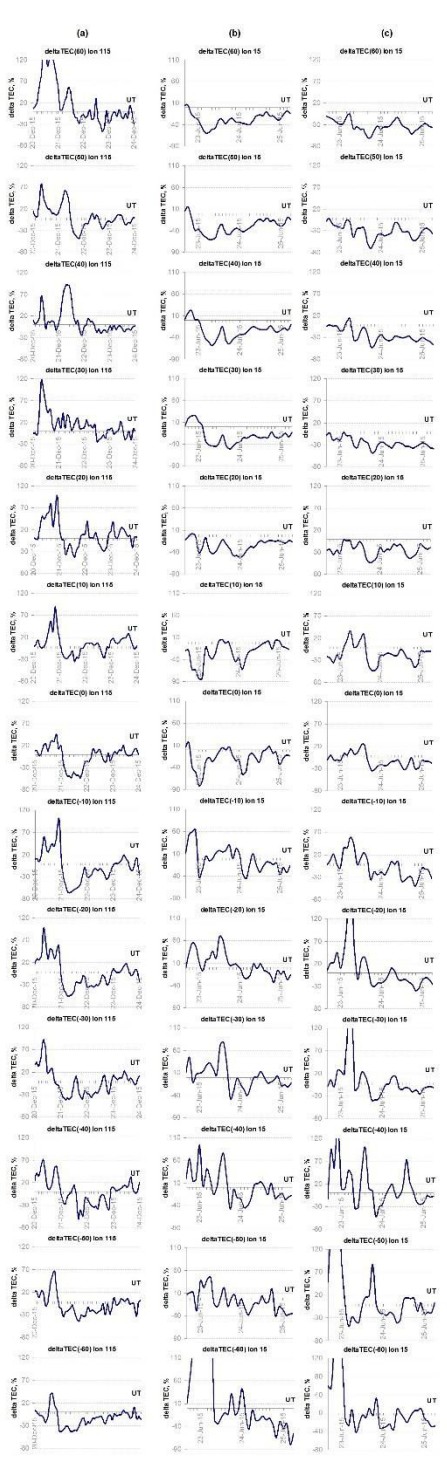

**Figure 3: δTEC variations between To and Te for storms: (a) #2 at Asian-Australian meridian; (b) #5 at Euro-African meridian; (c) #5 at Asian-Australian meridian.**

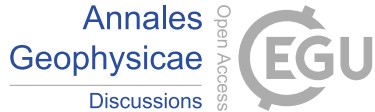





**Figure 4: The effect of δTEC bursts at all the latitudes between 60°N and 60°S along the longitude -100° (American sector) during the storm December 31st, 2015.**





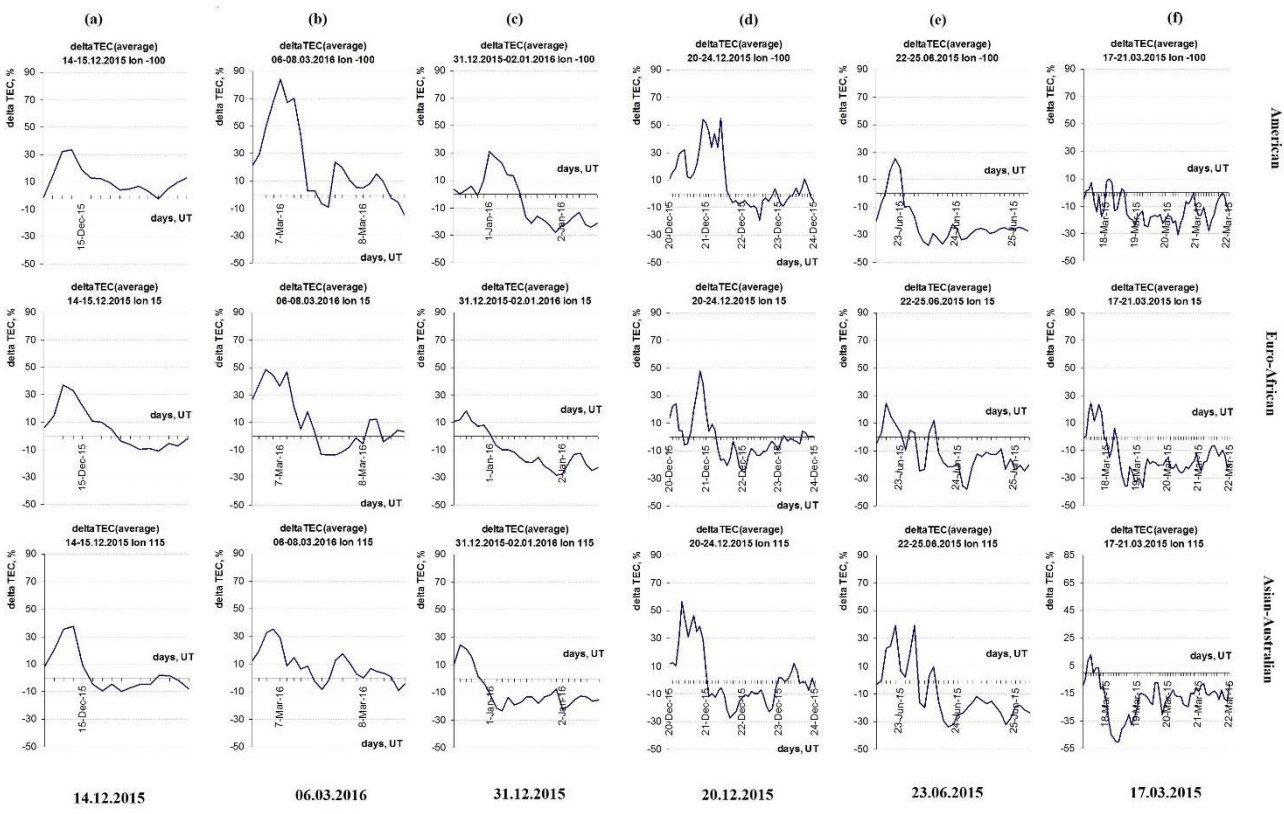

**Figure 5: Averaged along each meridian δTEC between To and Te.**





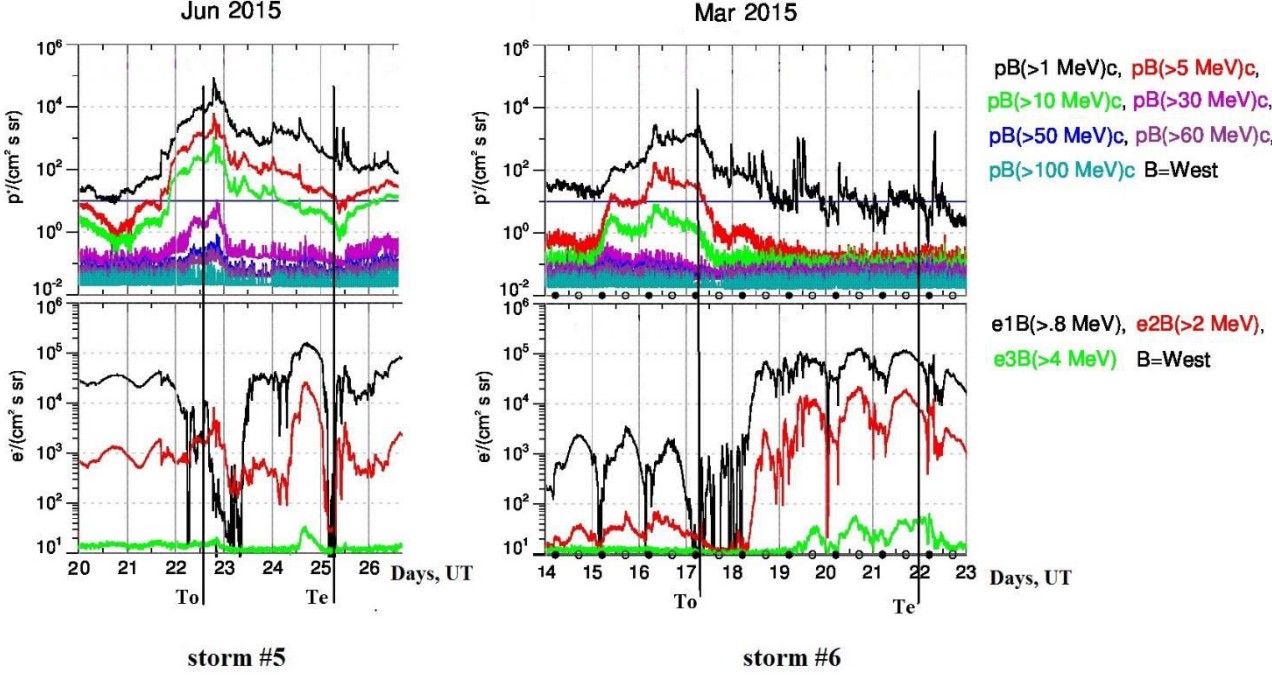

Figure 6: GOES satellite data for storms #5 and #6: p – protons, e – electrons. The particle energy is labeled by colors.