# Peer review of "Impact of magnetic storms on the global TEC distribution"

_Annales Geophysicae, 2018_

## Referee Comment (RC1) · Anonymous Referee #1 · 5 Mar 2018

The data and observations are interesting. The paper may become acceptable for publication after incorporating the following comments.

1. In addition to the Figures presented, I suggest adding some graphs with the TEC data observation during storm time period together with the average of the observations on quiet days with $\pm 1$ standard deviation.

2. In each graph from Figures 2 to 5, I suggest that the main and recovery phase of each geomagnetic storm be highlighted. For example, include a yellow and gray rectangle on each graph to represent the main and recovery phase of the sto

3. Figure 3: The resolution quality of this Figure is very poor.

4. I suggest that it be discussed, clearly, how each phase of the storms (main and

recovery phases) affect the ionosphere. The disturbances observed in the ionosphere during the storms were more pronounced during the main phase or recovery phase ???? Does the main and recovery phase affect the ionosphere in the same or different ways depending on the intensity of the storm ??? If necessary: a) include a new section to discuss only what was observed in the ionosphere during the main phase; b) subsequently do the same process for recovery phase.

---

## Referee Comment (RC2) · Anonymous Referee #2 · 30 Mar 2018

General Comments to the Authors:

The article is very interesting reporting significant findings. The results are of high quality and mostly well presented. However, there are some issues that need to be dealt with. These include the moving front, the inspection of actual TEC maps published by Madrigal Database, and the preferable usage of 1-min SYM-H index instead of the hourly Dst index. Based on the actual TEC maps, the description of moving front during the 31 December 2015 storm needs to be corrected.

Specific Comments:

Page (P) 3 Line (L) 7: "object", "subject" sounds better

P4 L10-15: As storms create sudden ionospheric and TEC changes, it

would be better to use actual TEC values provided by the Madrigal Database (http://cedar.openmadrigal.org/) than averaged 2-hourly GIM TEC values for storm studies. They use predictions to fill the data gaps and averaging over 2 hours that smooths out the storm induced sudden TEC variations.

P4 L25: The 1-min SYM-H index provided by the OMNI database (https://cdaweb.gsfc.nasa.gov/cdaweb/istp\_public/) and by the World Data Center for Geomagnetism, Kyoto (http://wdc.kugi.kyoto-u.ac.jp/aeasy/index.html) would be better than the 1-hour Dst index because of its higher time resolution. This higher resolution makes research more accurate. For example, the minimum SYM-H for the last storm, superstorm, was -234 nT reached at 2247 UT and not -223 nT at 2300 UT given by the hourly Dst index.

In Figure 3, the individual plots are too small and their labels are very hard to read.

P6 L30: I do not agree with the concept of disturbance front moving towards the equator applied to the 31 December 2015 storm. These TEC maps shown (see attached PDF) are from the Madrigal Database. The left column is for the end of 31 December 2015, the right column is for 3 hours latter. As the TEC maps show in the left column, there was a high TEC region in the American longitude sector and over the Pacific Ocean with large data gaps where GIM fills the gap with predicted values. But in these TEC maps, we can see actual TEC values and they show that these high TECs remained simultaneously at equatorial, low- and mid-latitudes. There was a peak over the magnetic equator, which is possibly the nighttime equatorial peak (or anomaly) implying that the vertical equatorial ExB drift was downward directed and drove a reverse plasma fountain that created this equatorial peak. However, at the same time, there were equally high TECs at mid-latitudes over the ocean. According to the velocity value given by the authors, the travelling time is 3 hours between +/- 40 GLAT and the equator. The right column shows the TEC maps 3 hours latter. As the storm progressed, we can see in the American sector the much lower TECs and the peaks of the Equatorial Ionization Anomaly (EIA) indicating that the vertical equatorial ExB drift was upward

ANGEOD
directed and drove a forward plasma fountain that created this EIA. The moving front section should be re-written and explained better because it is not supported by the actual TEC maps: there were equally high values at mid- and low-latitudes and over the equator (see left column). In terms of moving peaks, these actual TEC maps show that the equatorial peak turned into an EIA, characterized by a northern and a southern crest, as the vertical equatorial ExB drift flipped from downward to upward . So, the peak TEC moved from the equator to both hemispheres' lower latitudes and not from mid-latitude towards the equator as the authors claim.

P11 L25: As suggested, the authors should study the TEC maps of Madrigal Database regarding the moving front and make the necessary corrections.

P13 L10: Other data types (Dst/SYM-H, GOES) should be acknowledged as well.

Please also note the supplement to this comment: https://www.ann-geophys-discuss.net/angeo-2018-4/angeo-2018-4-RC2supplement.pdf

---

## Author Comment (AC1) · 21 May 2018

Responses to Anonymous Referee #1.

Anonymous Referee #1

The data and observations are interesting. The paper may become acceptable for publication after incorporating the following comments.

1. In addition to the Figures presented, I suggest adding some graphs with the TEC data observation during storm time period together with the average of the observations on quiet days with  $\pm 1$  standard deviation.

RESPONSE: We added examples of observed and median TEC values (see new Fig.

3a and Fig.3b). Median value serves as a quiet time reference.

2. In each graph from Figures 2 to 5, I suggest that the main and recovery phase of each geomagnetic storm be highlighted. For example, include a yellow and gray rectangle on each graph to represent the main and recovery phase of the sto

RESPONSE: We marked the main, recovery phases (MP and RP) and the end of the storms (Te) with vertical lines in new Figure 1 (left column), Figure 2, Figure 3, Figure 5, Figure 6.

3. Figure 3: The resolution quality of this Figure is very poor.

RESPONSE: The original source-file had a good quality but it was reduced when converting to pdf. In the new version of the manuscript we change the organization of the figure (Now it is Figure 4): now there are three panels (columns), each of which shows the results for the particular storm. Left plots of each panel display variations in the Northern Hemisphere and right plots– in the Southern Hemisphere.

4. I suggest that it be discussed, clearly, how each phase of the storms (main and recovery phases) affect the ionosphere. The disturbances observed in the ionosphere during the storms were more pronounced during the main phase or recovery phase ???? Does the main and recovery phase affect the ionosphere in the same or different ways depending on the intensity of the storm ??? If necessary: a) include a new section to discuss only what was observed in the ionosphere during the main phase; b) subsequently do the same process for recovery phase.

RESPONSE: We agree with the comment. The issue is discussed in the new version of the manuscript in detail.

WE THANK THE ANONYMOUS REFEREE #1 FOR HIS OR HER VALUABLE COM-MENTS ON OUR PAPER. WE ATTACH A NEW VERSION OF THE MANUSCRIPT TO THIS RESPONSE. THE CHANGES IN THE TEXT ARE IN BLUE FONT.

**ANGEOD**
Please also note the supplement to this comment: https://www.ann-geophys-discuss.net/angeo-2018-4/angeo-2018-4-AC1supplement.pdf

**ANGEOD**
Fig. 1.

**ANGEOD**

---

## Author Response (AR1)

**Responses to Anonymous Referee #1 (in blue)**

**Manuscript angeo-2018-4 "Impact of magnetic storms on the global TEC distribution" by Donat V. Blagoveshchensky et al.**

Anonymous Referee #1 Received and published: 5 March 2018

The data and observations are interesting. The paper may become acceptable for publication after incorporating the following comments.

1. In addition to the Figures presented, I suggest adding some graphs with the TEC data observation during storm time period together with the average of the observations on quiet days with±1 standard deviation.

We added examples of observed and median TEC values (see new Fig. 3a and Fig.3b). Median value serves as a quiet time reference.

2. In each graph from Figures 2 to 5, I suggest that the main and recovery phase of each geomagnetic storm be highlighted. For example, include a yellow and gray rectangle on each graph to represent the main and recovery phase of the sto We marked the main, recovery phases (MP and RP) and the end of the storms (Te) with vertical lines in new Figure 1 (left column), Figure 2, Figure 3, Figure 5, Figure 6.

3. Figure 3: The resolution quality of this Figure is very poor.

The original source-file had a good quality but it was reduced when converting to pdf.

In the new version of the manuscript we change the organization of the figure (Now it is Figure 4): now there are three panels (columns), each of which shows the results for the particular storm. Left plots of each panel display variations in the Northern Hemisphere and right plots– in the Southern Hemisphere.

4. I suggest that it be discussed, clearly, how each phase of the storms (main and recovery phases) affect the ionosphere. The disturbances observed in the ionosphere during the storms were more pronounced during the main phase or recovery phase???? Does the main and recovery phase affect the ionosphere in the same or different ways depending on the intensity of the storm ??? If necessary: a) include a new section to discuss only what was observed in the ionosphere during the main phase;
b) subsequently do the same process for recovery phase.
We agree with the comment. The issue is discussed in the new version of the manuscript in detail.

We thank the anonymous referee  $N \ge 1$  for his or her valuable comments on our paper. We attach a new version of the manuscript to this response. The changes in the text are in blue font.

**Responses to Anonymous Referee #2 (in blue)**

The article is very interesting reporting significant findings. The results are of high quality and mostly well presented. However, there are some issues that need to be dealt with. These include the moving front, the inspection of actual TEC maps published by Madrigal Database, and the preferable usage of 1-min SYM-H index instead of the hourly Dst index. Based on the actual TEC maps, the description of moving front during the 31 December 2015 storm needs to be corrected.

Specific Comments: Page (P) 3 Line (L) 7: "object", "subject" sounds better It was corrected.

P4 L10-15: As storms create sudden ionospheric and TEC changes, it would be better to use actual TEC values provided by the Madrigal Database (http://cedar.openmadrigal.org/) than averaged 2-hourly GIM TEC values for storm studies. They use predictions to fill the data gaps and averaging over 2 hours that smooths out the storm induced sudden TEC variations.

We would like to base our analysis on GIM TEC data. Two arguments can be given in favor to use GIM maps.

- TEC values obtained from Global Ionospheric Maps (GIM) used for this study proved their usefulness for estimation of TEC changes provoked by Space Weather events during decades (Hernandez-Pajares et al., 2009, doi: 10.1007/s00190-008-0266-1). For example, in (Sergeeva et al., 2017, doi: 10.1016/j.asr.2017.06.021; Sergeeva et al., 2018, doi: 10.4401/ag-7) it was shown that at the North American sector the difference between TEC obtained from GIM and TEC obtained from local RINEX data with high time resolution (3 min) is not significant.

- The differences between data from GIM and the Madrigal are not essential for the estimation of global variations considered in this paper.

P4 L25: The 1-min SYM-H index provided by the OMNI database

(https://cdaweb.gsfc.nasa.gov/cdaweb/istp\_public/) and by the World Data Center for Geomagnetism, Kyoto (http://wdc.kugi.kyoto-u.ac.jp/aeasy/index.html) would be better than the 1-hour Dst index because of its higher time resolution. This higher resolution makes research more accurate. For example, the minimum SYM-H for the last storm, superstorm, was -234 nT reached at 2247 UT and not -223 nT at 2300 UT given by the hourly Dst index.

New Figure 1 contains both indices of geomagnetic activity. SYM-H values were also added to new Table 1 to describe the storm with higher resolution.

As it is seen from Fig. 1 and also proved by other works (e.g. Wanliss and Showalter, 2006), there is no large difference between Dst and SYM-H to estimate the disturbance development. We marked the main and recovery phases of the storms with Dst-index. First, it was done for the illustrative purposes as Dst-curve is less rugged and, second, because we use classification of the intensity of the storms based on Dst-index.

The corresponding explanations were added to the text.

In Figure 3, the individual plots are too small and their labels are very hard to read.

The original source-file had a good quality but it was reduced when converting to pdf.

In the new version of the manuscript we change the organization of the figure (Now it is Figure 4): now there are three panels (columns), each of which shows the results for the particular storm. Left

plots of each panel display variations in the Northern Hemisphere and right plots- in the Southern Hemisphere.

P6 L30: I do not agree with the concept of disturbance front moving towards the equator applied to the 31 December 2015 storm. These TEC maps shown (see attached PDF) are from the Madrigal Database. The left column is for the end of 31 December 2015, the right column is for 3 hours latter. As the TEC maps show in the left column, there was a high TEC region in the American longitude sector and over the Pacific Ocean with large data gaps where GIM fills the gap with predicted values. But in these TEC maps, we can see actual TEC values and they show that these high TECs remained simultaneously at equatorial, low- and mid-latitudes. There was a peak over the magnetic equator, which is possibly the nighttime equatorial peak (or anomaly) implying that the vertical equatorial ExB drift was downward directed and drove a reverse plasma fountain that created this equatorial peak. However, at the same time, there were equally high TECs at mid-latitudes over the ocean. According to the velocity value given by the authors, the travelling time is 3 hours between +/-40 GLAT and the equator. The right column shows the TEC maps 3 hours latter. As the storm progressed, we can see in the American sector the much lower TECs and the peaks of the Equatorial Ionization Anomaly (EIA) indicating that the vertical equatorial ExB drift was upward directed and drove a forward plasma fountain that created this EIA. The moving front section should be re-written and explained better because it is not supported by the actual TEC maps: there were equally high values at mid- and low-latitudes and over the equator (see left column). In terms of moving peaks, these actual TEC maps show that the equatorial peak turned into an EIA, characterized by a northern and a southern crest, as the vertical equatorial ExB drift flipped from downward to upward. So, the peak TEC moved from the equator to both hemispheres' lower latitudes and not from mid-latitude towards the equator as the authors claim.

We thank the reviewer for the detailed comment. We will consider these issues carefully in our future analysis. In the new version of the manuscript we withdrew the section about the front moving as more similar cases are required to prove the results.

P11 L25: As suggested, the authors should study the TEC maps of Madrigal Database regarding the moving front and make the necessary corrections. The conclusion was withdrawn.

P13 L10: Other data types (Dst/SYM-H, GOES) should be acknowledged as well. It was done.

Please also note the supplement to this comment: https://www.ann-geophys-discuss.net/angeo-2018-4/angeo-2018-4-RC2supplement.pdf Interactive comment on Ann. Geophys. Discuss., https://doi.org/10.5194/angeo-2018-4, 2018.

We thank the anonymous referee  $N \ge 2$  for his or her valuable comments on our paper. We attach a new version of the manuscript to this response. The changes in the text are in blue font.

[revised manuscript text omitted]